# Biomimetic single Al-OH site with high acetylcholinesterase-like activity and self-defense ability for neuroprotection

Weiqing Xu[1], Xiaoli Cai[2], Yu Wu[1], Yating Wen[1], Rina Su[1], Yu Zhang[1], Yuteng Huang[2], Qihui Zheng[2], Liuyong Hu[3], Xiaowen Cui[4], Lirong Zheng[4], Shipeng Zhang[5], Wenling Gu[1], Weiyu Song ®[6], Shaojun Guo ®[5] ✉ & Chengzhou Zhu ®[1] ✉

Neurotoxicity of organophosphate compounds (OPs) can catastrophically cause nervous system injury by inhibiting acetylcholinesterase (AChE) expression. Although artificial systems have been developed for indirect neuroprotection, they are limited to dissociating P-O bonds for eliminating OPs. However, these systems have failed to overcome the deactivation of AChE. Herein, we report our finding that $Al^{3+}$ is engineered onto the nodes of metal–organic framework to synthesize MOF-808-Al with enhanced Lewis acidity. The resultant MOF-808-Al efficiently mimics the catalytic behavior of AChE and has a self-defense ability to break the activity inhibition by OPs. Mechanism investigations elucidate that $Al^{3+}$ Lewis acid sites with a strong polarization effect unite the highly electronegative –OH groups to form the enzyme-like catalytic center, resulting in superior substrate activation and nucleophilic attack ability with a 2.7-fold activity improvement. The multi-functional MOF-808-Al, which has satisfactory biosafety, is efficient in reducing neurotoxic effects and preventing neuronal tissue damage.

Acetylcholinesterase (AChE), as a vital hydrolase, modulates the normal neurotransmitter conduction for promoting cell development and maturation[1,2]. As the most toxic nerve agent, organophosphorus compounds (OPs) can strongly inhibit the activity of AChE and disrupt the nervous system function, leading to various violent diseases[3–7]. To alleviate the nerve damage, some organophosphorus hydrolase (OPH) mimics have been developed to preliminarily eliminate OPs and cooperate drugs to reactivate poisoned AChE for indirect neuroprotection[8–13]. However, the long-term deactivation of AChE is irreversible. Therefore, the design of AChE mimics that not only well perform the catalytic activity but also possess self-defense ability against OPs invasion essentially is greatly urgent.

In nature, ester hydrolysis is typical acid-base catalysis, which relies on the proton transfer process induced by the general acid/base function groups of enzymes[14]. First, the nucleophilic hydroxyl group (–OH) of serine attacks the carbon (phosphorus) atoms of substrates to form the intermediates. Meanwhile, the proximal basic imidazole groups and Lewis acid $Zn^{2+}$ sites respectively enrich the electron density of nucleophilic groups and improve the accessibility of carbon (phosphorus) atoms[15,16]. Subsequently, the dissociation of the ester bond occurs and generates the first product with the assistance of protonation[17,18]. Although some enzyme mimics with hydrolase-like activity have been developed through effective screening of metal active sites[19–21], most of them are limited to catalyzing the dissociation

[1]National Key Laboratory of Green Pesticide, International Joint Research Center for Intelligent Biosensing Technology and Health, College of Chemistry, Central China Normal University, Wuhan 430079, P.R. China. [2]Department of Nutrition, Hygiene and Toxicology, School of Public Health, Medical College, Wuhan University of Science and Technology, Wuhan 430065, P.R. China. [3]School of Materials Science and Engineering, Wuhan Institute of Technology, Wuhan 430205, P.R. China. [4]Beijing Synchrotron Radiation Facility, Institute of High Energy Physics Department, Chinese Academy of Sciences Institution, Beijing 100049, P.R. China. [5]School of Materials Science and Engineering, Peking University, Beijing 100871, P.R. China. [6]State Key Laboratory of Heavy Oil Processing, College of New Energy and Materials, China University of Petroleum, Beijing 102249, P.R. China. ✉e-mail: guosj@pku.edu.cn; czzhu@ccnu.edu.cn

of highly reactive phosphate esters rather than carboxylic esters hydrolysis due to their insufficient activity. The rational design of advanced AChE mimics requires vivid mimicking of the catalytic pocket of enzymes, which involves not only strong Lewis acid sites ($M^{n+}$) but also highly electronegative nucleophilic groups (−OH) for synergistic catalysis.

Herein, we design and tune the bioinspired Lewis acid sites in metal−organic frameworks (MOFs) for boosting the AChE-like activity and self-defense ability for neuroprotection (Fig. 1). $Al^{3+}$ was decorated onto the Zr-oxo clusters in MOF-808 to form the new and accessible Al−OH sites with enhanced Lewis acid activity via a simple post-synthetic cluster metalation strategy. The AChE-like activity of the resultant MOF-808-Al is superior to the pristine MOF-808. Importantly, it conquers the activity inhibition by OPs owing to the self-defense and even detoxification ability. Mechanistic studies reveal that the Lewis $Al^{3+}$ sites have a stronger polarization effect than the $Zr^{4+}$ sites, strengthening the electrophilicity and reactivity of C atoms. Assisted by the highly electronegative −OH groups, the proposed MOF-808-Al exhibits a decreased energy change for the dissociation of ester bonds and desorption of hydrolysates, leading to a 2.7-fold activity improvement. The efficient expression of AChE-like activity of MOF-808-Al in vitro is confirmed. We show that the MOF-808-Al with good biocompatibility is efficient in reducing OP-induced neurotoxic effects to alleviate apoptosis and tissue injury, realizing effective neuroprotection.

## Results

### Synthesis and characterization of hydrolase mimics

The MOF-808 was exposed to aluminum nitrate solution at 85 °C for 6 h to introduce high Lewis acidity $Al^{3+}$ (Fig. 2a). In this typical metalation process, terminal groups ($-OH_2/OH$) of Zr-O nodes were deprotonated, and serve as the anchored sites to incorporate the heterometal cations[22,23]. The powder X-ray diffraction (XRD) result reveals the well-retained crystal structure of MOF-808 upon introducing Al (Supplementary Fig. 1)[24]. The resultant MOFs share a similar regular octahedral structure (~200 nm), revealed by scanning electron microscopy (SEM) and transmission electron microscopy (TEM)

results (Fig. 2b, Supplementary Fig. 2). All the elements are uniformly distributed throughout the whole MOF-808-Al and without discernible nanoparticles, evidenced by the high-angle annular dark-field scanning TEM (HAADF-STEM) and corresponding energy dispersive X-ray spectroscopy (EDS) mapping, and high-resolution TEM images (Fig. 2c, Supplementary Fig. 3). The introduction of Al species makes the micropores dominated and reduces the pore volume of MOFs from 0.74 cm g$^{-1}$ to 0.63 cm g$^{-1}$ (Fig. 2d), and the Brunauer−Emmett−Teller surface area from 1438.6 m$^2$ g$^{-1}$ to 1284.3 m$^2$ g$^{-1}$ (Supplementary Fig. 4), which reveal that additional Al species were incorporated into the mesopores and micropores of MOF. To probe the precise location of Al species, diffuse reflectance infrared Fourier transform spectroscopy (DRIFTS) was conducted. As can be seen in Fig. 2e, two typical peaks at 3670.4 and 2745.3 cm$^{-1}$ observed in MOF-808 are assigned to $-OH_2/OH$ and $\mu_3-OH$ groups of Zr-O clusters, respectively[25,26]. After introducing $Al^{3+}$, the apparent decrease in intensity of both peaks implies the interaction between the external $Al^{3+}$ and these groups. X-ray photoelectron spectroscopy (XPS) spectra for O *1s* reveal that the M−O characteristic peak of MOF-808-Al shifts to higher binding energy (Fig. 2f), and its relative content is larger than that of the MOF-808, suggesting the coexistence of Al-O species[27]. In addition, Fourier transform infrared (FTIR) spectroscopy analysis shows the MOF-808-Al with additional signal peaks at 1017.0, 2154.1, and 583.3 cm$^{-1}$, assigned to Al−OH and Zr−O−Al bonds, respectively (Fig. 2g, Supplementary Fig. 5)[28,29]. The atomic ratio of Zr to Al was calculated as 7.4 by inductively coupled plasma optical emission spectrometer, implying that ~0.81 Al atom was immobilized onto the per $Zr_6O$ cluster. Taken together, the modification of heterometal cations conforms to the metalation mechanism, where the $Al^{3+}$ coordinates with the Zr−O node to form Zr−O−Al sites.

The effect of $Al^{3+}$ on the chemical environment of Zr sites, which is closely correlated to the Lewis acidity[30], was investigated by X-ray absorption spectroscopy (XAS) and XPS. The Zr K-edge X-ray absorption near-edge structure (XANES) spectra show that the absorption edge of MOF-808-Al is reproduced fairly well with the MOF-808 and close to $ZrO_2$ (Fig. 3a, Supplementary Fig. 6), verifying that the valence state of Zr in both MOFs is +4. After coordination with Zr−O nodes, the oxidation state of Al remains unaltered at +3, evidenced by XPS analysis (Supplementary Fig. 7). Fourier transform-extended X-ray absorption fine structure (EXAFS) scillations present no apparent change after chemical modification with $Al^{3+}$, explaining that the structure of $Zr_6O$ cluster is negligibly influenced (Fig. 3b). The wavelet transform (WT) of Zr K-edge EXAFS was conducted to better discriminate the overlapped contribution of various elements. As illustrated in Fig. 3c, compared with MOF-808, the WT map of MOF-808-Al appears the additional signal near the Zr−Zr location (in difference spectra, marked by a red dashed line), attributed to the weak scattering of Zr−Al path[31]. These results demonstrate that $Al^{3+}$ is attached to the second shell of the $Zr_6O$−core through terminal groups instead of embedded within it. On the one hand, the well-retained electronic and coordinated structure of $Zr^{4+}$ in MOF-808-Al suggests that the addition coordination structure (Al−OH) does not influence the electronegativity and accessibility of $Zr^{4+}$ sites, maintaining the inherent acidity. On the other hand, the terminal $Al^{3+}$ is expected to serve as an extra Lewis acid site for improving the catalytic performance.

The Lewis acidity of MOFs was quantitatively determined by fluorescence and colorimetric spectroscopy. First, a common fluorescent dye N-methylacridone (NMA), features lone pairs that can combine with exposed Lewis metal sites, resulting in a redshift in its fluorescence maxima ($\lambda_{max}$)[32]. As illustrated in Fig. 3d, MOF-808-Al exhibits a larger redshift than pristine MOFs, demonstrating its higher Lewis acidity. Apart from NMA, electron-rich superoxide ions free radicals ($O_2^{\cdot-}$) with skyscraping reactivity can strongly bind with electrophilic Lewis acid sites to form $M(O_2^{\cdot-})$ species, confirmed by electron paramagnetic resonance (EPR) spectroscopy (Supplementary

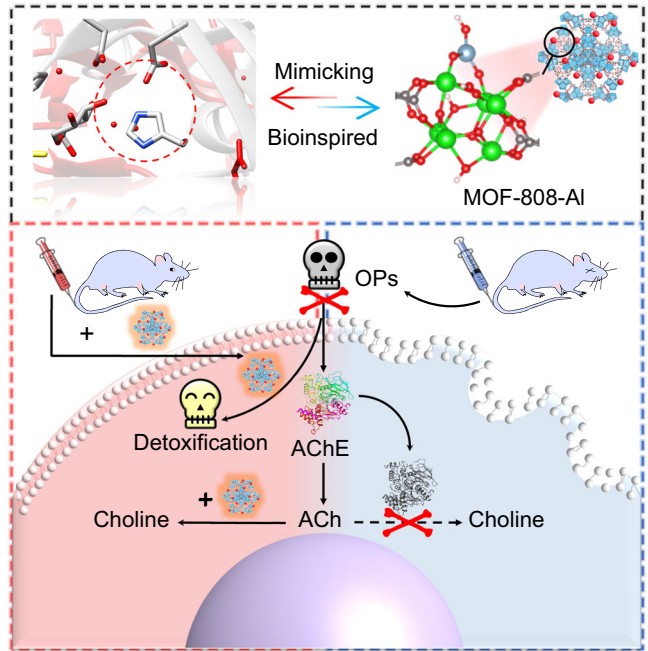

**Fig. 1 | Scheme of neuroprotection strategy by MOF-808-Al.** Schematic illustration of the bioinspired active center of MOF-808-Al, and it serves as the AChE mimic against OPs for neuroprotective effect.

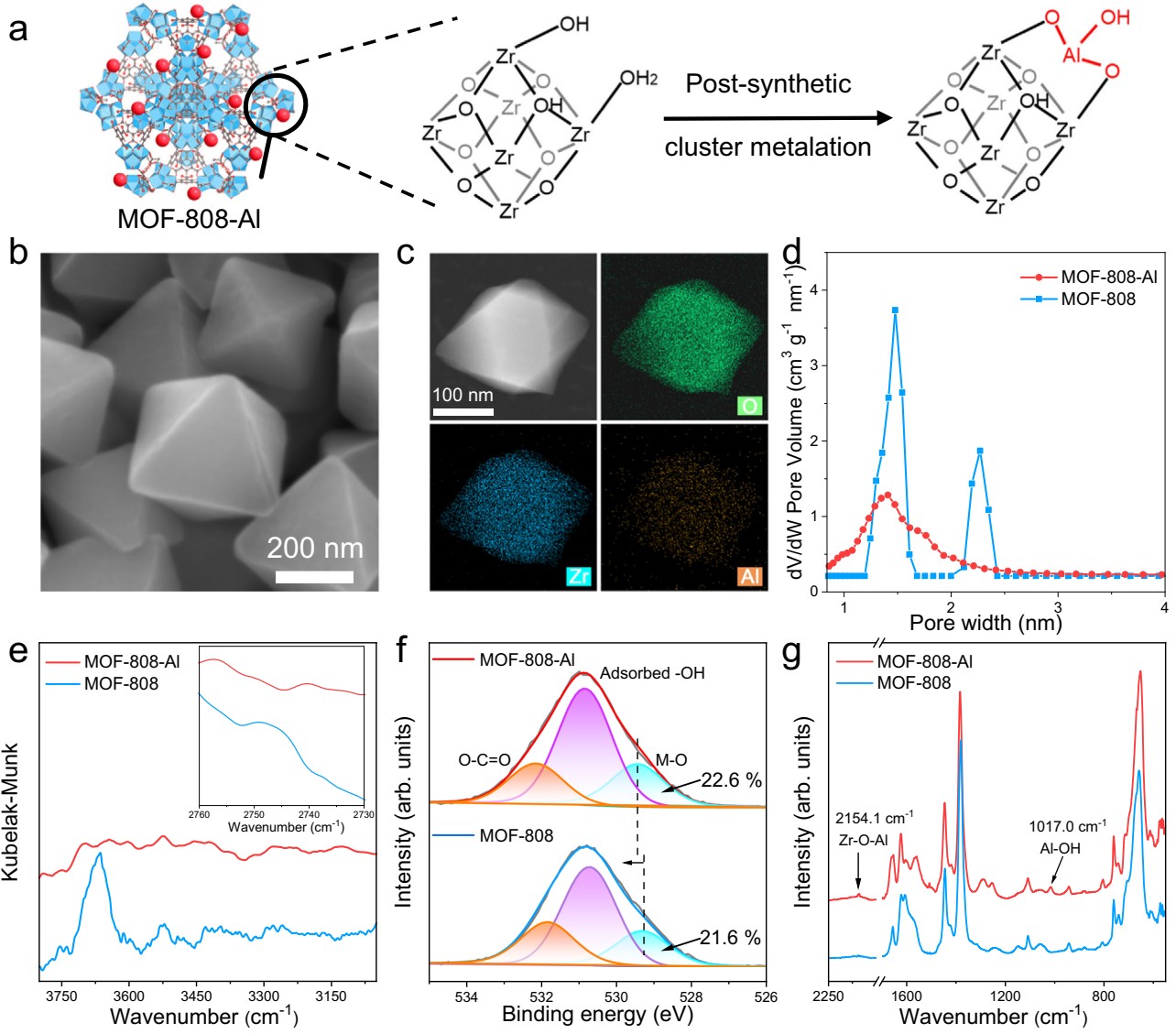

**Fig. 2 | Synthesis and characterizations of MOFs. a** Schematic representation of the synthesis of MOF-808-Al. **b** SEM and **c** HAADF-STEM and corresponding EDS mapping images of MOF-808-Al. The experiments were repeated three times with similar results. **d** pore size distribution, **e** DRIFTS spectra (*Inset*: detailed spectra for the highlighted bond around 2745.3 cm⁻¹), **f** O *1s* XPS spectra, and **g** FTIR spectra of MOFs.

Fig. 8)[33]. Here, a typical nitro blue tetrazolium (NBT)-$O_2^{\cdot-}$ colorimetric reaction was performed. In detail, $O_2^{\cdot-}$ can reduce the NBT into blue formazan and show an observable absorption signal[34]. The formation of $M(O_2^{\cdot-})$ species will suppress this colorimetric reaction (Supplementary Fig. 9). As a result, the amounts of MOFs are linearly related to the relative decrease in absorbance (at 662 nm), and the slope of the MOF-808-Al system is 2.1-fold larger than that of the MOF-808 (Fig. 3e). It quantitatively confirms the increased Lewis acidity after introducing $Al^{3+}$. Given that the consumption of terminal −OH groups in the metalation process, which is bound to decrease the amount of Zr−OH site, the $CD_3CN$ infrared experiment was conducted to characterize the relative content of acid sites (Supplementary Fig. 10). As summarized in Fig. 3f, more than two times higher Lewis acid sites by $Al^{3+}$ doping are further confirmed. Although the relative content of M−OH sites is reduced after the modification of $Al^{3+}$, it is still larger than that of Lewis acid sites, which is adequate to achieve synergistic catalysis.

### Mimicking enzymes for AChE and OPs hydrolysis

Under the alkaline condition, both MOFs that serve as the substitute of AChE were introduced into the acetylcholine (ACh) hydrolysis system. The mass spectrum and FTIR analysis reveal that the obtained MOFs can perform the function of AChE to hydrolyze ACh into choline, verifying their AChE-like properties (Supplementary Figs. 11 and 12). To quantify the catalytic activity, a multienzyme cascade reaction was conducted (Supplementary Fig. 13), where the hydrolysate choline can be cascade catalyzed by choline oxidase (ChOx) and horseradish peroxidase (HRP) to oxidize colorimetric substrates (3,3',5,5'- tetramethylbenzidine, TMB)[35]. As shown in Fig. 4a, both MOFs can trigger the cascade reaction, and MOF-808-Al processes a higher hydrolytic activity than MOF-808. Besides, the doping of Al significantly boosts the reaction rate of MOFs (Supplementary Fig. 14). Under optimal conditions (pH 9.0) (Supplementary Fig. 15), the kinetic experiments reveal that the MOF-involved reaction conforms to the Michaelis–Menten equation (Fig. 4b). In comparison with MOF-808, MOF-808-Al has a lower Michaelis constant ($K_M$), demonstrating its better affinity ability for ACh. Consequently, the catalytic efficiency ($k_{cat}/K_M$) of MOF-808-Al is 2.7 times higher than that of MOF-808 (Supplementary Table 1). The more favorable hydrolytic kinetics is believed to be closely related to the increased

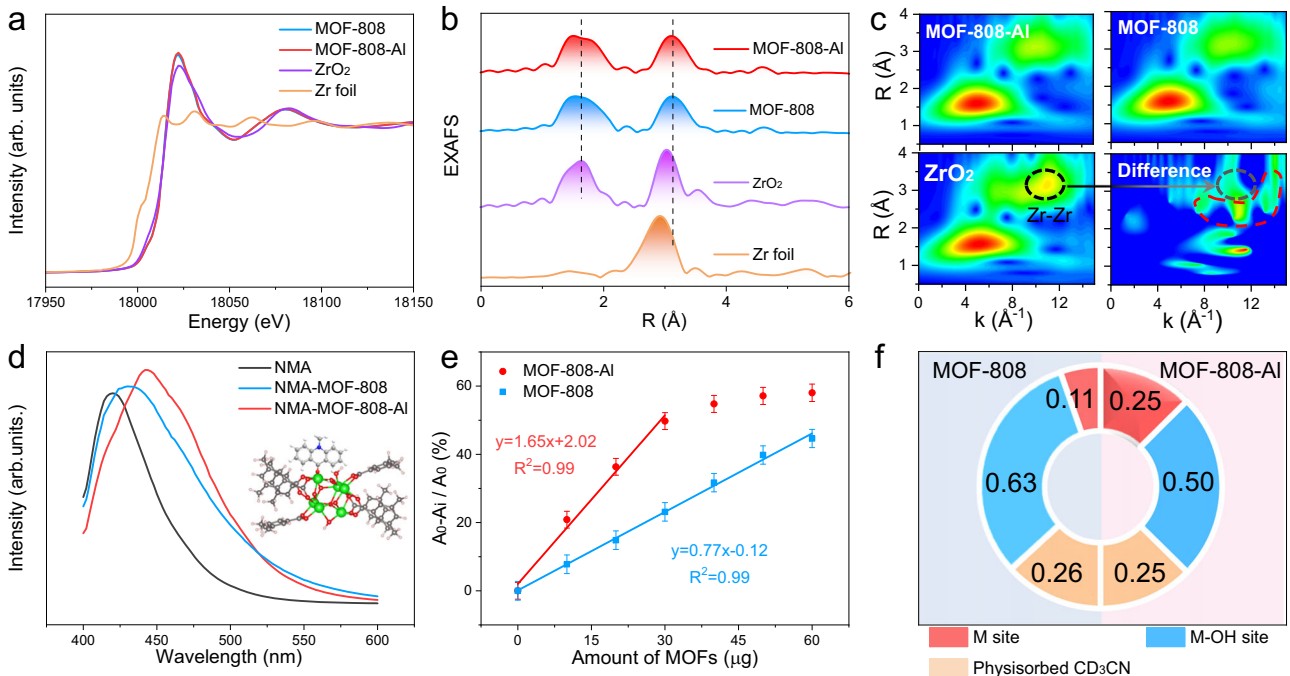

**Fig. 3 | Chemical structure and Lewis acidity of catalysts. a** Zr K-edge XANES and **b** Fourier transform-EXAFS spectra of MOFs, Zr foil, and ZrO$_2$. **c** WT for the K$^3$-weighted EXAFS signals of MOFs, ZrO$_2$, and difference spectra between MOF-808-Al and MOF-808. **d** Fluorescence spectra of NMA, NMA-MOF-808, and NMA-MOF-808-Al composites (inset: illustration of the coordination of NMA to the MOF). **e** The relative decreased absorbance of formazan in the presence of different amounts of MOFs ($A_O$ and $A_i$ are the absorbance values of reaction systems without MOFs and with different amounts of MOFs, respectively). Error bars mean ± standard deviations (s.d.) calculated from three independent measurements. **f** A pie graph showing the relative contents of M/M–OH sites and physisorbed CD$_3$CN in MOFs. The gray indicates MOF-808 (left), pick indicates MOF-808-Al (right).

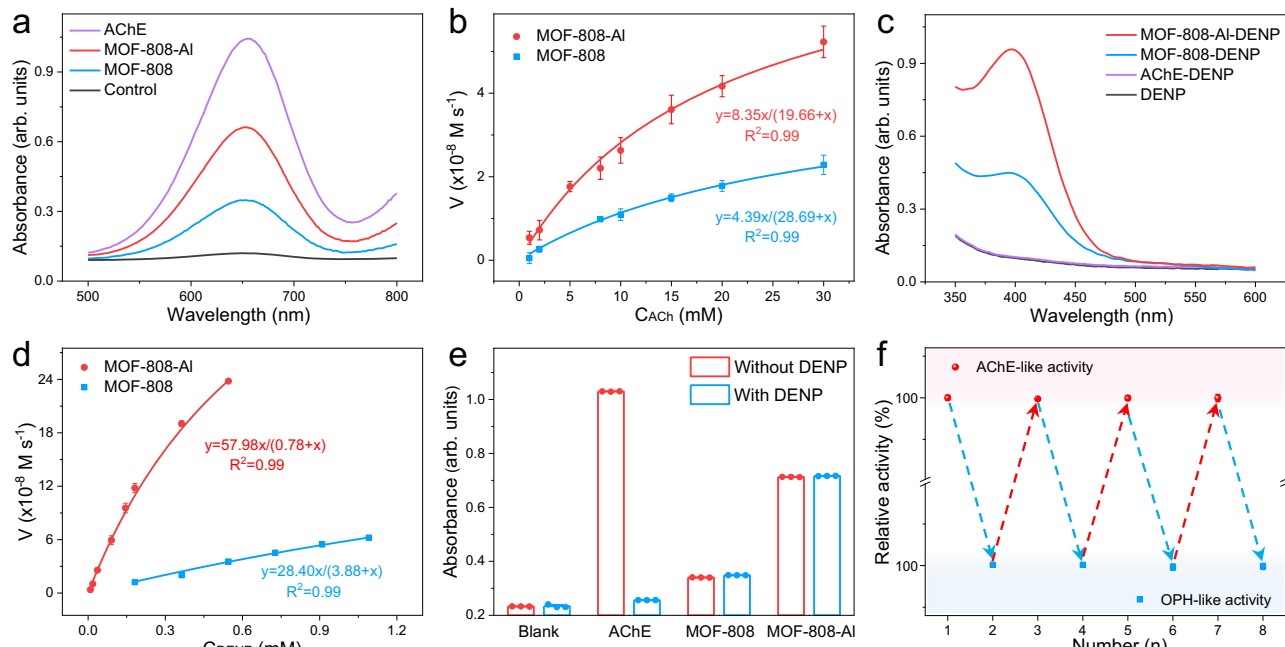

**Fig. 4 | Enzyme-like performance of catalysts. a** Absorption spectra of ACh hydrolysis reaction catalyzed by AChE, MOFs in the presence of ChOx and HRP. **b** The kinetic curve of both MOFs toward ACh. Error bars mean ± s.d. calculated from three independent measurements. **c** Absorption spectra of DENP hydrolyzed by three catalysts, including AChE, MOF-808, MOF-808-Al. **d** The kinetic curve of both MOFs toward DENP. Error bars mean ± s.d. calculated from three independent measurements. **e** Absorbance values (at 652 nm) of ACh hydrolysis reaction catalyzed by AChE and MOFs before and after treatment with DENP. Error bars mean ± s.d. calculated from three independent measurements. **f** Relative activity of ACh and DENP hydrolysis reaction catalyzed by MOF-808-Al after cycle several times. Error bars mean ± s.d. calculated from three independent measurements. The gray indicates OPH-like activity (bottom), pick indicates AChE-like activity (top).

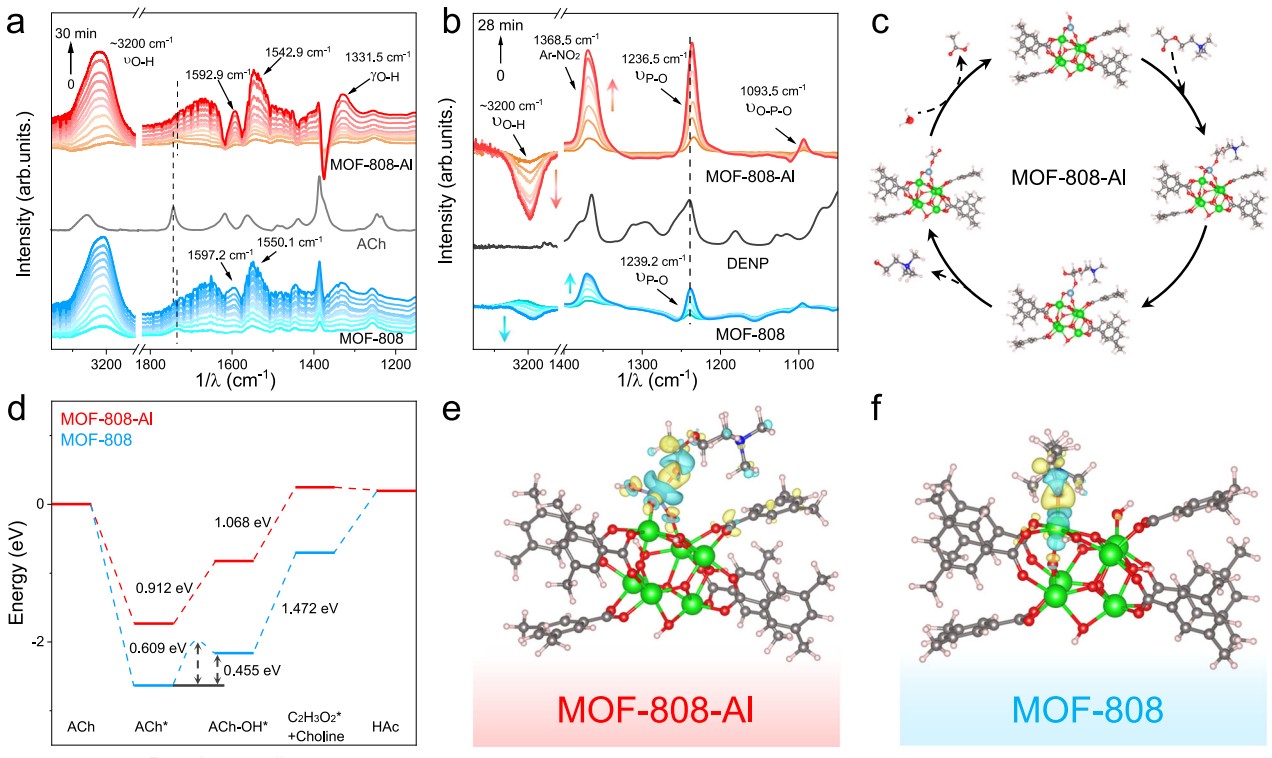

**Fig. 5 | Reactive intermediates monitoring and DFT calculations.** In situ ATR-FTIR spectra of MOF-induced **a** ACh and **b** DENP hydrolysis reactions. **c** Proposed reaction process on MOF-808-Al. **d** The energy diagram of MOF-808 and MOF-808-Al system. **e**, **f** Calculated charge density differences study the bonding interactions among Zr/Al, O, and C atoms and the charge transfer (yellow/blue isosurfaces denote an increase/decrease of electron density).

Lewis acid sites in MOF-808-Al, promoting the adsorption and activation of substrates. Apart from ACh, a simulant acetylthiocholine (ATCh) was also employed as the hydrolytic substrate. The superior catalytic efficiency of MOF-808-Al further confirms the enhanced AChE-like property of MOF-808-Al (Supplementary Figs. 16 and 17, and Supplementary Table 2). Moreover, the AChE-like activity of resultant MOFs is well retained after treatment with high temperatures and small organic molecules due to the inherent stability of nanomaterials (Supplementary Fig. 18). In addition to the good AChE-mimicking property, MOF-808-Al can effectively degrade the organophosphate-based nerve agent simulant (such as diethyl-4-nitrophenylphosphate, DENP) (Fig. 4c, Supplementary Figs. 19 and 20)[36]. In contrast, AChE fails to hydrolyze DENP because of its inherently remarkable specificity. The kinetic experiments show that the hydrolytic efficiency ($K_{cat}/K_M$) of MOF-808-Al toward DENP (1923.08 $M^{-1}$ $min^{-1}$) is more than one order of magnitude larger than that of MOF-808 (188.14 $M^{-1}$ $min^{-1}$) (Fig. 4d and Supplementary Table 3). The inherent OPH-like activity of MOFs encourages us to investigate their tolerability toward OPs. On the one hand, the AChE-like ability of MOFs is well retained after pretreatment with DENP for 0.5 h (Fig. 4e). XRD and XPS results reveal that the crystal structure and chemical valence of the obtained mimics are unchanged after treatment with DENP, indicating high structural stability (Supplementary Figs. 21 and 22). While the catalytic performance of the AChE shows an obvious decrement owing to the structure destruction by the phosphorylation of DENP. On the other hand, after repeatedly introducing DENP in the hydrolytic system several times (Fig. 4f), MOF-808-Al not only almost keeps the initial AChE-like activity but also can self-degrade and eliminate the invasive OPs. Benefiting from these, we believe that the proposed intelligent AChE mimics integrate self-defense and detoxification abilities to efficiently break the activity inhibition by OPs, possessing a huge promise in practice (Supplementary Fig. 23).

## Insights into the underlying hydrolysis mechanism

FTIR spectroscopy was performed to study the interactions between the ACh and MOFs. As can be seen in Supplementary Fig. 24, the C=O vibration (1746.1 $cm^{-1}$) of the ester group shifts to a lower frequency upon mixing with MOFs. It is believed that the carbonyl is adsorbed on the Lewis metal sites, resulting in the increased polarization of the double bond[24]. Based on this, the *operando* attenuated-total-reflection FTIR (ATR-FTIR) spectroscopy was conducted for in situ monitoring of the hydrolytic process. As illustrated in Fig. 5a, the same shift of the C=O absorption band is observed at the initial reaction stage. Besides, the carbonyl signal intensity is significantly reduced, and the carboxyl anions signals (1592.9 and 1542.9 $cm^{-1}$) are increased over time, implying the hydrolysis of the ester group. Notably, these vibration frequencies are lower than in the MOF-808 system (1597.2 and 1550.1 $cm^{-1}$), revealing that the additional $Al^{3+}$ site provides a stronger interaction force to induce the polarization of the C=O bond. Correspondingly, two prominent absorption bands at ~3200 and 1331.5 $cm^{-1}$, assigned to the −OH vibration of the generated choline and acetic acid, increase gradually. The higher polarization effect of MOF-808-Al for DENP hydrolysis was also confirmed, where the signal peaks of P=O are shifted to the lower frequency (1236.5 $cm^{-1}$) relative to the MOF-808 system (1239.2 $cm^{-1}$) (Fig. 5b). The absorption band of nitrobenzene (1368.5 $cm^{-1}$) appears and increases over time, verifying the generation of hydrolysate p-nitrophenol. It should be pointed out that significant consumption of peak M−OH (~3200 $cm^{-1}$) is observed in the MOF-808-Al system at the initial reaction and then reaches equilibrium. The reaction efficiency of MOF-808-Al is higher than that of MOF-808. It reveals that, apart from the Lewis acid sites, the terminal −OH groups serve as another catalytic moiety to perform the hydrolysis. To further

verify the synergistic effect between the M−OH sites and Lewis acid sites, a traditional Lewis acid, $Al_2O_3$ nanomaterial with negligible M-OH site, was employed to hydrolyze substrates[37]. As can be seen in Supplementary Fig. 25, the alone $Al_2O_3$ presents poor hydrolase-like properties, which is enhanced by the introduction of the MOF-808. The activity of the integrated system is higher than that of the accumulation of $Al_2O_3$ and MOF-808. According to previous reports[38,39], it is believed that the exposed $M^{n+}$ ($Al^{3+}$ or $Zr^{4+}$) combines with oxygen of carbonyl groups to increase the electrophilicity and reactivity of carbon atoms by polarization effect. Subsequently, the cleavage of the C−O bond occurs under the nucleophilic attack of neighboring −OH, being accompanied by the generation of choline with the help of the protons transfer. For the third step, the intermediate is further hydrolyzed to acetic acid in the presence of $H_2O$, along with the regeneration of catalysts (Supplementary Fig. 26). As for DENP hydrolysis, a similar catalytic process occurs, except for the activation of the P=O bond (Supplementary Fig. 27).

Theoretical calculations were implemented to further explore the underlying mechanisms for the enhanced AChE-like performance. At first, MOF-808-Al and MOF-808 cluster model structures were established and optimized (Supplementary Fig. 28). Different from the pristine MOF-808, $Al^{3+}$ with terminal −OH group is seated on the Zr−O clusters and acted as the catalytic sites in the MOF-808-Al system to perform the hydrolysis (Fig. 5c, Supplementary Fig. 29). As can be seen in Fig. 5d, the energy change for nucleophilic attack of Zr−OH is calculated to be 0.455 eV, which is lower than that of the Al−OH (0.912 eV). Additionally, the transition state search indicates that the barrier of the Zr−OH attack is 0.609 eV. However, this process in the MOF-808-Al system is controlled by thermodynamics. Although the attack of Zr−OH* exhibits energetically favorable, the energy change is lower than that of the breaking of the C=O bond and the formation of choline, which is the rate-determining step (RDS). Compared with the MOF-808 system (1.472 eV), the lower energy change of the MOF-808-Al system (1.068 eV) indicates higher hydrolytic activity. Furthermore, the possibility of adjacent Zr−OH* for the nucleophilic attack in the MOF-808-Al system was examined (Supplementary Fig. 30). It is revealed that the nucleophilic attack of Zr−OH* is RDS with an energy change of 1.23 eV. This process is less favorable in comparison with the Al-OH* attack. Therefore, it is believed that the carbon atom is most likely to be attacked by Al-OH* in the MOF-808-Al system. We speculate the enhanced activity is related to the stronger polarization effect of $Al^{3+}$ sites. In this regard, the charge density difference analysis was conducted to study the interaction between MOFs and ACh. When adsorbed on metal sites, an observable electron migration occurs in the C=O bond of ACh, reducing the electron density of C atoms (Fig. 5e, f). Furthermore, the Bader charge calculation shows that the C atom, upon conjugation to Al sites has a larger positive charge than that of the conjugation to Zr sites (Supplementary Fig. 31). The increased electropositivity of C atoms is helpful for the subsequent nucleophilic attack. Notably, the valence electron of nucleophilic O atom on Al sites (7.60) is higher than the Zr sites (7.29) after adsorption of the substrates. This phenomenon is attributed to the fact that the substrate serves as the electron donor[40], enhancing the electronegativity of −OH groups. In short, the obtained MOF-808-Al, with the improved polarization effect and nucleophilic ability, achieves efficient hydrolysis.

## Cytoprotection performance

Here, the *neuroendocrine cell line* PC12 was selected as an experimental model to evaluate the protective effect. The cell cytotoxicity experiment was first carried out by MTT (3-(4,5- dimethylthiazol-2-yl)-2,5-diphenyltetrazolium bromide) assay, revealing the good biosafety of MOF-808 and MOF-808-Al (Fig. 6a, Supplementary Fig. 32). The same result is obtained by the cell live/dead double staining assay (Supplementary Fig. 33). Both MOFs show hydrolytic activity toward ACh and

DENP at pH 7.4 (Supplementary Figs. 15 and 34). Subsequently, the expression of AChE-like activity of the obtained mimics was investigated in intracellular. Compared with the neat PC12 cells, MOF-808-Al-involved groups show higher hydrolytic activity and concentration dependence (Fig. 6b). Impressively, after introducing DENP, the expression of AChE activity in PC12 cells is significantly decreased, while the MOF-808-Al-involved groups still maintain appreciable activity. Besides, the inhibition effect alleviates with the increase of the amounts of MOF-808-Al, implying its detoxification. The AChE-like property of MOF-808 was also confirmed in vitro, while its inhibition effect is lower than that of MOF-808-Al due to insufficient catalytic activity (Supplementary Fig. 35). The influence of invasive DENP on mitochondrial membrane potential was investigated by 5′,6,6′-tetrachloro−1,1′,3,3-tetraethylbenzimidazolylcarbocyanine iodide (JC-1) fluorescent probe[41]. As illustrated in Fig. 6c and Supplementary Fig. 36, the presence of unamiable DENP results in an obvious reduction in the red fluorescent intensity of J aggregates ($E_m = 595$ nm) and an increase in green fluorescent intensity of the JC-1 monomer state ($E_m = 530$ nm), implying the unbalanced membrane potential. The destruction of mitochondrial function results in apoptosis, as evidenced by MTT experiments (Supplementary Fig. 37). Fortunately, this damage is alleviated with the aid of MOF-808-Al. Furthermore, the corresponding confocal laser scanning microscopy (CLSM) imaging was carried out. As shown in Fig. 6d, e, DENP-poisoned cells present abnormal green fluorescent signals. In contrast, the MOF-808-Al-involved system displays strong red fluorescence before and after being treated with DENP. Therefore, it could be concluded that the proposed MOF-808-Al featuring self-defense and detoxification functions, can realize satisfactory cytoprotection with good biosafety. Given the OP-induced oxidative stress injury reported by previous studies[42,43], the protective effect of MOF-808-Al was further evaluated. The content of reactive oxygen species (ROS) in DENP-treated PC12 cells was determined by a common fluorescent probe (2′,7′-dichlorodihydrofluorescein diacetate, DCFH-DA) (Supplementary Fig. 38). The invasion of DENP significantly increases the level of ROS, which is effectively reduced with the treatment of the MOF-808-Al, being equivalent to that of the normal cells. This result verifies that the detoxification function of MOF-808-Al can attenuate the DENP-induced excessive ROS production and further oxidative stress damage. Apart from this, because the metabolic abnormality of ACh is closely related to neuropsychiatric disorders, a MOF-808-Al-based biosensor was constructed to monitor the ACh concentration. As depicted in Supplementary Fig. 39, the sensitive determination of ACh was achieved with a limit of detection of 1.03 μM, showing a good DENP anti-poisoning ability (Supplementary Table 4). Additionally, the MOF-808-Al exhibits higher signal output intensity than MOF-808, implying its better performance for monitoring intracellular ACh (Supplementary Fig. 40).

## In vivo therapeutic effects

The efficient cytoprotective activity in vitro and good biosafety encouraged us to study the therapeutic effect of MOF-808-Al on OP-induced damage in vivo. The mouse model was established by injecting DENP, MOF-808-Al, and MOF-808-Al + DENP into the tail vein for 8 consecutive days (Fig. 7a)[44]. As shown in Fig. 7b, the weight loss of the DENP-treated group implies the successful onset of the OP-poisoned model. Noteworthily, the presence of MOF-808-Al significantly alleviates the decrease in body weight loss. Besides, the activity of AChE in the serum of mice was detected to estimate the potential for neurotoxic responses induced by OPs (Fig. 7c). As for the DENP-treated group, the AChE expression is inhibited compared to the control group. With the help of MOF-808-Al, the AChE activity is recovered, which indicates the efficient ability for alleviating the AChE damage, agreeing with the intracellular experiments. The brain and histology changes of mice were tested by using the hematoxylin−eosin (HE) staining. As can be seen in Fig. 7d, Supplementary Fig. 41, the

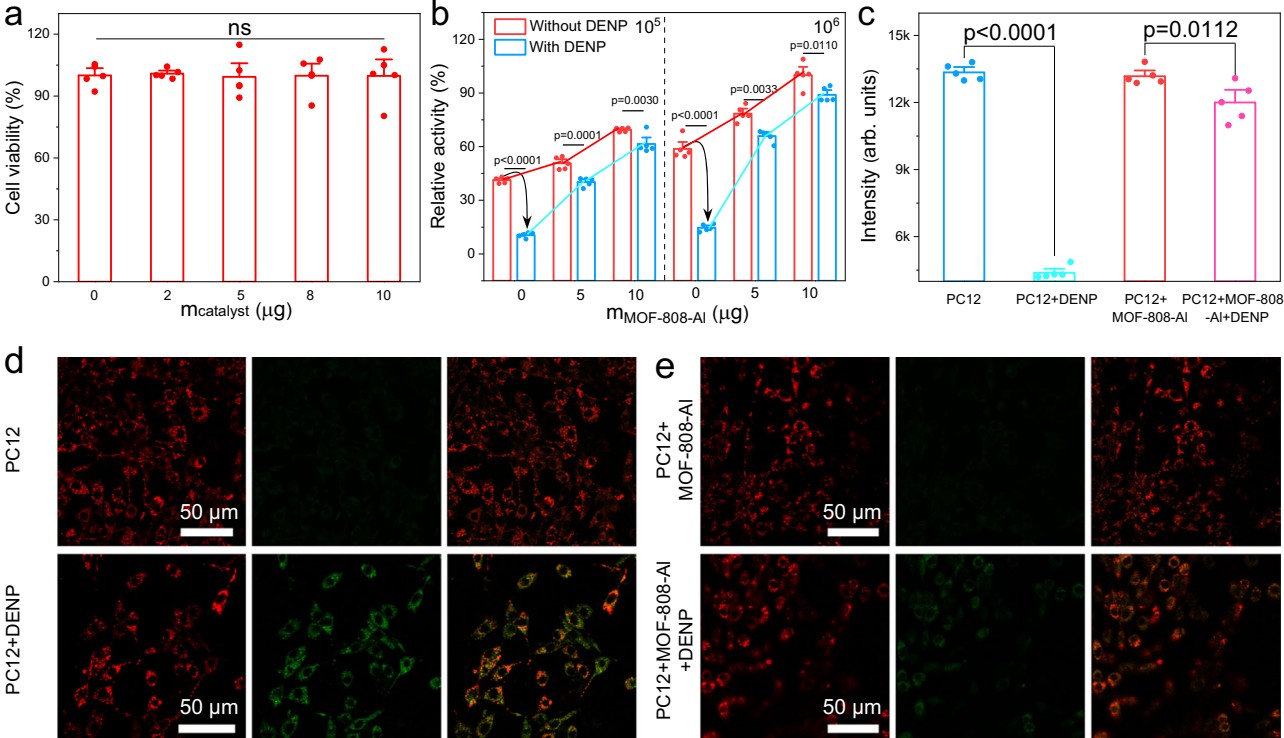

**Fig. 6 | In vitro cellular evaluations. a** Cell viability after treatment with different contents of MOF-808-Al for PC12 cells. Data are represented as mean ± s.d. (*n* = 5 independent experiments, ns represented no statistical difference). **b** Relative activity of AChE in PC12 cells before and after treatment with DENP in the presence of various amounts of MOF-808-Al. Data are represented as mean ± s.d. (*n* = 5 independent experiments). **c** Fluorescence intensity of JC-1 ($E_m$: 595 nm) in different systems, including PC12, PC12 + DENP, PC12 + MOF-808-Al, and PC12 + MOF-808-Al + DENP, and **d, e** corresponding cell images showing fluorescence changing. Data are represented as mean ± s.d. (n = 5 independent experiments). Statistical significance is assessed by the Student's two-tailed test. Source data are provided in the Source Data file.

hippocampus, and main organs, including the heart, lung, liver, spleen, and kidney in the MOF-808-Al group show no observable abnormity, further confirming the favorable biocompatibility. However, upon the introduction of DENP, the cellular morphology in the hippocampus presents significant change. Besides, granulovacuolar dystrophy and cytoplasmic vacuolization are observed in the liver and kidney (Fig. 7e). Notably, these damages can be effectively ameliorated with the assistance of MOF-808-Al. Additionally, the levels of three ROS-involved biomarkers in the liver tissues of mice, including superoxide dismutase (SOD), glutathione (GSH), and malonaldehyde (MDA), were evaluated. The elevated MDA level and decreased expression of SOD and GSH induced by DENP are closely related to excessive ROS production (Fig. 7f–h)[45]. With the aid of MOF-808-Al, the expressions of these biomarkers in DENP-treated tissues are the same as the normal tissues. Furthermore, the alkaline phosphatase (ALP) expression in serum, which is usually regarded to be associated with liver disorders, was tested (Fig. 7i). The increased level of ALP conferred by DENP was reduced as much as in healthy mice with the assistance of MOF-808-Al. Besides, as a control, the MOF-808-Al-treated group exhibits normal expression of biochemical markers. Therefore, the proposed AChE mimics with good biocompatibility exhibit a good protective effect in alleviating neurotoxic effects and inducted tissue injuries.

## Discussion

In summary, we report highly Lewis acidity Al³⁺ decorated Zr-O nodes of MOF-808 as intelligent AChE mimics with high self-defense ability against OPs for efficient neuroprotective effect. Mechanistic studies elucidate that the improved property of MOF-808-Al stems from the stronger polarization effect of Lewis Al³⁺ sites and the higher electron density of −OH groups than that of the Zr⁴⁺ sites. The synergistic effect between the Al³⁺ and −OH is favorable for substrate activation and subsequent nucleophilic attack, thereby promoting the cleavage of ester bonds and desorption of hydrolysates. Due to the inherent good stability and degradation ability toward OPs, the proposed AChE mimics break the neurotoxic poisoning, alleviating apoptosis and neuronal tissue damage. We expect that the design principle of this work can be leveraged to develop other advanced enzyme mimics by vividly mimicking the catalytic pocket of enzymes.

## Methods

### Synthesis of MOF-808-Al

The as-prepared MOF-808 (100 mg) was dispersed in 40 mL DMF containing 0.1 M Al(NO₃)₃·9H₂O and placed in a round bottom flask. The mixture was heated to 85 °C for 6 h with continuous stirring. The solid was obtained by centrifuging (9500×*g*, 3 min) and washed with DMF and acetone three times. Finally, through a solvent exchange process in the acetone for 2 days, MOF-808-Al was prepared by vacuum drying at 100 °C.

### Evaluation of AChE-like activity of MOFs

The activity of AChE was determined by integrating ChOx and HRP cascade reactions. Briefly, ACh can be hydrolyzed by AChE to generate acetic acid and choline, and the latter oxidized by ChOx to produce H₂O₂, which catalyzes the TMB with the help of HRP. The obtained absorbance value (at 652 nm) is positively correlated with the AChE activity. Totally, 2 mg mL⁻¹ MOFs (10 μL) or AChE (50 mU mL⁻¹, 10 μL) and 1 mM ACh (50 μL) were added into the 0.1 M HEPES buffer solution (pH 9.0, 500 μL), and were co-incubated for 20 min. The supernatant (150 μL) was obtained by centrifugation, and 1 U mL⁻¹ ChOx (10 μL), 50 μg mL⁻¹ HRP (10 μL), and TMB (1 mM, 130 μL) were introduced.

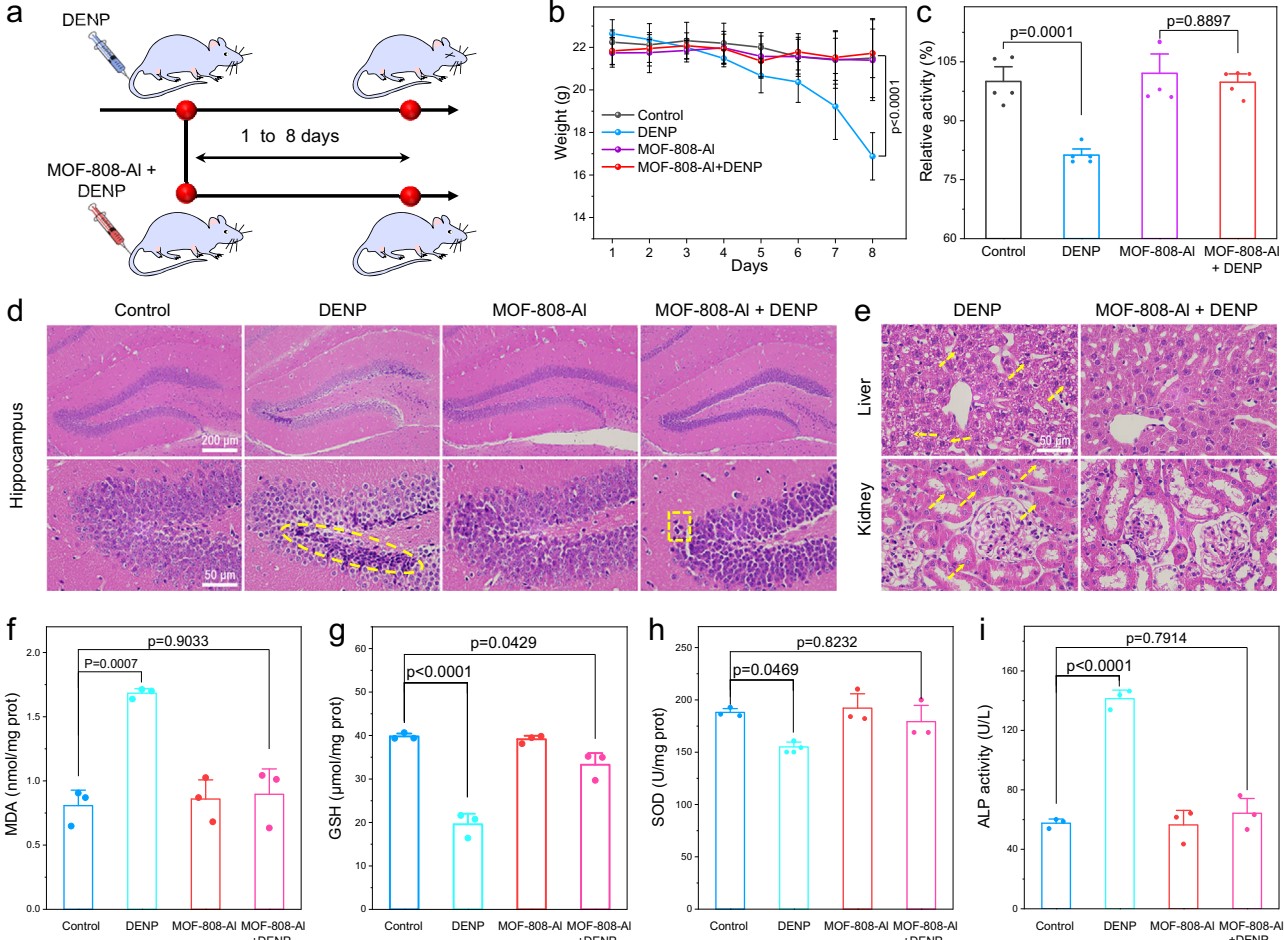

**Fig. 7 | In vivo protective effect. a** Schematic illustration of the OP-damaged model mice and the MOF-808-Al-mediated treatment. **b** Mice weight after treatment with saline, DENP, MOF-808-Al, and MOF-808-Al + DENP for 8 days. Data are represented as mean ± s.d. ($n$ = 4 independent mice). **c** Relative activity of AChE in the serum of mice after various treatments. Data are represented as mean ± s.d. ($n$ = 4 independent serum of mice). H&E staining **d** hippocampus and **e** liver and kidney after different treatments. The level of **f** MDA, **g** GSH, **h** SOD in liver tissues of mice, and **i** the level of ALP in the serum of mice after different treatments. Data are represented as mean ± s.d. ($n$ = 3 independent serum of mice). Statistical significance is assessed by the Student's two-tailed test. Source data are provided in the Source Data file.

After incubation for another 8 min, the absorption spectra were recorded using the microplate reader for further analysis.

## Monitoring the hydrolytic process of MOFs by in-situ FTIR

In total, 10 mg mL$^{-1}$ MOFs (8 μL) was first dropped on the diamond internal reflection element (IRE) and immobilized with 2 μL 0.05 mM Nafion. The HEPES buffer solution (0.8 mL, pH 9.0) was added, and the FTIR spectrum was recorded as a reference. Then, 100 mM ACh was further introduced, and the spectra were recorded every 3 min for further analysis. As a control, the spectrum of the ACh was recorded.

As for monitoring the hydrolysis of DENP, the same procedures were performed, except that the ACh was replaced by the 5 mM DENP, and the spectra were recorded every 4 min.

## Activity assay of intracellular AChE

PC12 cells were from the American Type Culture Collection (ATCC)] CRL-1721.1) and seeded into 96-well plates at several densities of cells (10$^5$ and 10$^6$) per well (200 μL) and were cultured at 37 °C with 5% CO$_2$ for 24 h. Different amounts of MOF-808-Al were added and incubated for 24 h. Then, 10 μL of DENP (0.5 μM, PBS buffer) was introduced and incubated for 6 h. Subsequently, the medium solution was removed, and 20 mM ACh (50 μL) and 0.1 M HEPES buffer solution (pH 9.0, 100 μL) were added and incubated for 30 min. Then, 1 U mL$^{-1}$ ChOx

(10 μL), 50 μg mL$^{-1}$ HRP (10 μL), and TMB (1 mM, 130 μL) were further introduced and co-incubated for another 8 min. The absorption spectra were recorded by the microplate reader.

## Protective effect of MOF-808-Al in vivo

The healthy C57BL/6 mice (4 weeks) were purchased from the Hubei Provincial Center for Disease Control and Prevention. The in vivo animal experiments were performed according to the guidelines of the National Institutes of Health Guide for the Care and Use of Laboratory Animals and approved by the Ethics Committee of Medical College, Wuhan University of Science and Technology (2023082). The mice were raised with free access to standard feed and water in a 12 h dark–light cycle and the ambient conditions of room temperature (20–24 °C), 50 ± 5% relative humidity. A total of 16 mice were randomly assigned to 4 cages ($n$ = 4 for every group). One cage of mice was used as a negative control and treated with only normal saline. Three cages of mice were separately treated with DENP (10 μg kg$^{-1}$), MOF-808-Al (2 mg kg$^{-1}$), and MOF-808-Al + DENP every day. After treatment for 8 days, all mice were anesthetized using diethyl ether. Blood was collected and centrifuged for 15 min at 860×$g$ to collect serum samples for further biochemical analysis. Tissues (brain, kidney, heart, lung, liver and spleen) from each mouse were dissected and immediately frozen in liquid nitrogen and stored at −80 °C.

## Histological analysis

The main organs and hippocampus from the control group and treatment groups were fixed in 10% formalin, embedded in paraffin wax, sectioned at 4 μm thickness, and stained with H&E for microscopic observation.

## Reporting summary

Further information on research design is available in the Nature Portfolio Reporting Summary linked to this article.

## Data availability

The data supporting the findings of this study are available within the article and its Supplementary Information files. All other data are available from the corresponding author(s) upon request. Source data are provided in this paper.

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

## Acknowledgements

C.Z. gratefully acknowledges the financial support of the start-up fund from the Central China Normal University, National Natural Science Foundation of China (no. 22074049), the Fundamental Research Funds for the Central Universities (No. CCNU22JC006), and the Program of Introducing Talents of Discipline to Universities of China (111 programs, B17019). We thank the 1W1B station in Beijing Synchrotron Radiation Facility (BSRF) for X-ray absorption spectroscopy measurements.

## Author contributions

C.Z. and S.G. supervised the research. W.X. designed and performed most of the synthesis and characterizations. X.C., Y.H., and Q.Z. carried out the in vivo experiments. X.W.C. and L.Z. conducted the X-ray absorption fine structure characterization. Y.W., L.H., S.Z., W.G., and W.S. supported data analysis. W.S. supported the DFT calculation. W.X. (lead) and all other authors (supporting) wrote the paper. All authors discussed the results and assisted during paper preparation.

## Competing interests

The authors declare no competing interests.
