## [Peer Review File · Nature Communications]

Reviewers' Comments:

Reviewer #1:

Remarks to the Author:

Guo, Zhu and col. report on the Al³⁺ doping of MOF-808 system and its catalytic hydrolytic behavior towards organophosphoester and acetylcholine hydrolysis. Most of the work on metal-organic framework materials has been focused on the capture and degradation organophosphate toxic compounds. The reported approach is innovative. However, there are a series of issues that need to be addressed before this contribution is suitable for publication.

1) The structural characterization is in agreement with the attachment of the Al³⁺ to the Zr₆ cluster instead of the actual exchange with Zr⁴⁺ ion. The synergistic effect on the hydrolytic activity of the MOF-808 system has been previously shown with Lewis acid Mg²⁺, however, in the previous report the results suggest that Mg²⁺ is exchanging one Zr⁴⁺ center in the cluster (ref. 18). Having a look to the isotherm shape it seems that there is a certain loss of the mesoporous structure indicating that Al³⁺ incorporation might take place at the mesopores (see ref. 18). If the Zr to Al ratio is 7.2 should it not be the Al content 0.83 per Zr₆ cluster?

2) A second part of the study is devoted to the catalytic hydrolytic activity of the Al³⁺ doped MOF-808 system. All the reported activity seems to have been carried out at pH 9.0 (HEPES buffer). Since the main focus of the study is a biomedical application for organophosphate poisoning treatment (organophosphate hydrolysis) and AChE like activity of the MOF-808-Al system the working pH should be 7.4. In this regard, some MOF systems are not hydrolytically active/become poisoned at lower pHs. The actual behavior of the system in both organophosphate and acetylcholine hydrolysis need to be confirmed at pH 7.4.

3) The following references on dual detoxification by capture/hydrolysis of organophosphate and reactivation of acetylcholine esterase activity are relevant to the current study ACS Applied Materials & Interfaces, 2022, 14, 26501-26506 and Inorg. Chem. 2023, 62, 13, 5049-5053

There are certain parts of the manuscript that are difficult to understand.

Reviewer #2:

Remarks to the Author:

The manuscript entitled "Biomimetic Single Al-OH Site with High Acetylcholinesterase-Like Activity and Self-Defense Ability for Neuroprotection" reported a novel multifunctional MOF-808-Al with highly Lewis acidity Al³⁺-decorated Zr-O nodes as AChE mimics against organophosphate compounds. Moreover, the proposed AChE mimics exhibit protective effect in alleviating neuronal tissue damage. In general, a series of characterization and theoretical calculation methods were used to verify the catalytic activity of the enzyme-like catalytic center (Al-OH) site and the self-defense ability to breaking the neurotoxic poisoning. This work provides a mechanistic explanation to the structure-activity relationship at the atomic level and it is undoubtedly of high value for those working in this specialized field of designing tunable advanced enzyme mimics and related applications.

However, I still feel that the innovative aspect of this paper is not highlighted enough to warrant publication in Nature Communications. Below are my main concerns with this paper.

(1) In the mitochondrial membrane potential assay, the fields of view for CLSM imaging in Fig 5d and 5e are almost identical and the authors did not explain how different treated groups capture the same vision.

(2) Methods for evaluating the therapeutic effects in vivo should be describe in paper body. In Fig 6c, the activity of AChE was detected in the blood from the mouse or other samples? More experimental evidence is required to validate the protective effect except HE staining analysis.

(3) In addition to MTT assay, biocompatibility and safety evaluation of MOF-808-Al in vivo should be carried out.

(4) There is no scale bar in Figure 1b, Fig 5d and 5e. Last but not least, a small spelling error in introduction (To alleviate the never damage) but there are several like this in the manuscript so

the language needs carefully polishing.

Reviewer #3:

Remarks to the Author:

Xu et al. report a biomimetic heterogeneous MOF-808-Al catalyst with the ability to act like acetylcholinesterase, a critical enzyme required for the functioning of the nervous system. Importantly, this catalyst also decomposes neurotoxic organophosphate compounds. They show that the catalyst is therefore a good treatment for organophosphate (OP) poisoning in cell lines and in mice. This work is exciting and significant as while enzymes have been explored for OP poisoning, heterogeneous catalysts have not.

I feel that the work would be suitable for publication after major revisions addressing the below points:

1) The evidence for the catalytic mechanism of action and why the addition of Al^{3+} increases the activity of MOF-808 is not clearly presented. In Figure 4d, the authors present a reaction energy diagram to compare the energy profile of acetylcholine hydrolysis catalyzed by MOF-808 and MOF-808-Al. However, the authors only calculated the energies of the intermediate states, and not the transition states. The authors should clarify how they obtained energy barriers of 1.472 eV for the MOF-808 system and 1.068 eV for the MOF-808-Al system, as stated on page 8.

2) The authors postulate that the role of Al^{3+} is to act as a strong Lewis acid to activate acetylcholine, and to increase the nucleophilicity of OH^* , which is required for attacking the conjugated acetylcholine to hydrolyze it. This is supported by Bader charge analyses showing a larger positive charge on the carbonyl C after conjugation to Al^{3+} than conjugation to Zr^{4+} , and a larger negative charge on OH^* groups attached to Al^{3+} than to Zr^{4+} .

While these are reasonable arguments, they should be supported by transition state calculations to calculate the barriers of the attack of OH^* on the carbonyl C, i.e. the $ACh^* \rightarrow ACh-OH^*$ step in Figure 4d. For MOF-808-Al, acetylcholine and OH^* are both attached to the same Al, which means the transition state likely goes through a 4-membered ring state, which may be energetically unfavorable. On the other hand, for MOF-808, acetylcholine and OH^* are attached to different Zr centers, and so the TS may be energetically more favourable. For MOF-808-Al, the authors may also consider attack of a nearby $Zr-OH^*$ on the Al-ACh* complex, which may be more favorable.

3) The authors point out that M-OH* groups are depleted during the reaction of MOF-808-Al with DENP (Figure 4b), which means that the reaction is not catalytic as the MOF catalyst will become deactivated after all M-OH* groups are consumed. Are M-OH* groups depleted during the reaction of the MOF catalysts with acetylcholine as well? If one OH^* group can only react with one molecule of acetylcholine/DENP, would this mean that high doses of the catalyst would be needed to achieve protection/neutralization of OP poisoning, and would these doses be reasonable for typical poisoning cases in humans?

Minor points:

1) The Bader charge figure (Figure S29) is confusing as the numbers in the figures are the absolute number of valence electrons, instead of the charge, as is claimed in the caption.

2) Could the authors explain why MOF-808 was chosen for functionalization, instead of simple ZrO_2 nanoparticles for example?

3) The authors should explain how the "Free Energies" in Figure 4d were obtained—were frequency calculations also performed with VASP?

Reply to reviewer 1:

Guo, Zhu and col. report on the Al^{3+} doping of MOF-808 system and its catalytic hydrolytic behavior towards organophosphoester and acetylcholine hydrolysis. Most of the work on metal-organic framework materials has been focused on the capture and degradation organophosphate toxic compounds. The reported approach is innovative. However, there are a series of issues that need to be addressed before this contribution is suitable for publication.

Q1: The structural characterization is in agreement with the attachment of the Al^{3+} to the Zr_6 cluster instead of the actual exchange with Zr^{4+} ion. The synergistic effect on the hydrolytic activity of the MOF-808 system has been previously shown with Lewis acid Mg^{2+} , however, in the previous report the results suggest that Mg^{2+} is exchanging one Zr^{4+} center in the cluster (ref. 18). Having a look to the isotherm shape it seems that there is a certain loss of the mesoporous structure, indicating that Al^{3+} incorporation might take place at the mesopores (see ref. 18). If the Zr to Al ratio is 7.2 should it not be the Al content 0.83 *per* Zr_6 cluster?

R1: Thanks for your valuable comment. In this work, a typical post-synthetic metalation strategy was used to introduce heterometal (Al^{3+}) to the Zr_6 -core. In details, the terminal $-\text{H}_2\text{O}$ groups and $\mu_3\text{-OH}$ groups as secondary building units in Zr-MOFs have a pK_a of 3–4, and can be deprotonated to bind other metal centers (**Fig. R1**) (*Angew. Chem. Int. Ed.* **2015**, *54*, 14696–14700; *J. Am. Chem. Soc.* **2021**, *143*, 8829–8837). To confirm the precise location of the Al species within the MOF, a series of characterization methods, including diffuse reflectance infrared Fourier transform spectroscopy (DRIFTS), X-ray photoelectron spectroscopy (XPS), and X-ray absorption spectroscopy (XAS), were employed. Note the coordination structure and valence state of Zr in MOFs present no apparent change after the Al^{3+} modification. We believe the model of ZrO-core with attached Al^{3+} is suitable to describe MOF-808-Al in this work. However, as for the previous work (*J. Am. Chem. Soc.* **2019**, *141*, 11801–11805, ref. 18), the coordination number of the Zr-O cluster was reduced after introducing Mg^{2+} , which can be attributed to the formation of $\text{Mg}_x\text{Zr}_{6-x}$ heterometallic clusters.

Fig. R1. Illustration of the incorporation of heterometal into the clusters; red indicates the deprotonated ligands (*Angew. Chem. Int. Ed.* **2015**, *54*, 14696–14700).

As shown in **Fig. 1d**, the introduction of Al species makes the micropores dominated and reduces the pore volume of MOFs from $0.74 \text{ cm}^3 \text{ g}^{-1}$ to $0.63 \text{ cm}^3 \text{ g}^{-1}$. Therefore, it is believed that additional Al species were incorporated into the mesopores and micropores of MOF. According to your nice suggestion, the relative expression of the distribution of Al species is updated in the revised manuscript. In addition, the content of Al in the ZrO cluster is corrected.

“The introduction of Al species makes the micropores dominated and reduces the pore volume of MOFs from $0.74 \text{ cm}^3 \text{ g}^{-1}$ to $0.63 \text{ cm}^3 \text{ g}^{-1}$ (**Fig. 1d**), and the Brunauer-Emmett-Teller (BET) surface area from $1438.6 \text{ m}^2 \text{ g}^{-1}$ to $1284.3 \text{ m}^2 \text{ g}^{-1}$ (**Supplementary Fig. 4**), which reveal that additional Al species were incorporated into the mesopores and micropores of MOF.”

Q2: A second part of the study is devoted to the catalytic hydrolytic activity of the Al^{3+} doped MOF-808 system. All the reported activity seems to have been carried out at pH 9.0 (HEPES buffer). Since the main focus of the study is a biomedical application for organophosphate poisoning treatment (organophosphate hydrolysis) and AChE like activity of the MOF-808-Al system the working pH should be 7.4. In this regard, some MOF systems are not hydrolytically active/become

poisoned at lower pHs. The actual behavior of the system in both organophosphate and acetylcholine hydrolysis need to be confirmed at pH 7.4.

R2: According to your valuable suggestion, the hydrolytic performance of both MOFs for acetylcholine (ACh) and organophosphate (diethyl-4-nitrophenylphosphate, DENP) was evaluated at pH 7.4. As shown in **Figs. R2** and **R3**. The MOF-808-AI shows a higher hydrolytic behavior toward ACh and DENP than those of MOF-808. Relative experiments are updated in the revised manuscript.

Fig. R2. Absorbance values (at 652 nm) of ACh hydrolysis reaction in different pH buffer solutions.

Fig. R3. Absorbance values (at 405 nm) of DENP hydrolysis reaction in different pH buffer solutions.

Q3: The following references on dual detoxification by capture/hydrolysis of organophosphate and reactivation of acetylcholinesterase activity are relevant to the current study ACS Applied Materials & Interfaces, 2022, 14, 26501-26506 and Inorg. Chem. 2023, 62, 13, 5049–5053.

R3: According to your valuable suggestion, these works are cited in the revised manuscript.

“To alleviate the nerve damage, some organophosphorus hydrolase (OPH) mimics have been developed to preliminarily eliminate OPs and cooperate drugs to reactivate poisoned AChE for indirect neuroprotection⁸⁻¹³.”

Q4: There are certain parts of the manuscript that are difficult to understand.

R4: According to your nice suggestion, this manuscript was further revised very carefully.

Reply to reviewer 2:

The manuscript entitled “Biomimetic Single Al-OH Site with High Acetylcholinesterase-Like Activity and Self-Defense Ability for Neuroprotection” reported a novel multifunctional MOF-808-Al with highly Lewis acidity Al³⁺ decorated Zr-O nodes as AChE mimics against organophosphate compounds. Moreover, the proposed AChE mimics exhibit protective effect in alleviating neuronal tissue damage. In general, a series of characterization and theoretical calculation methods were used to verify the catalytic activity of the enzyme-like catalytic center (Al-OH) site and the self-defense ability to breaking the neurotoxic poisoning. This work provides a mechanistic explanation to the structure-activity relationship at the atomic level and it is undoubtedly of high value for those working in this specialized field of designing tunable advanced enzyme mimics and related applications.

However, I still feel that the innovative aspect of this paper is not highlighted enough to warrant publication in Nature Communications. Below are my main concerns with this paper.

Q1: In the mitochondrial membrane potential assay, the fields of view for CLSM imaging in Fig 5d and 5e are almost identical and the authors did not explain how different treated groups capture the same vision.

R1: Thanks for your valuable comment. The relative CLSM images have been corrected in the revised manuscript and are given below. As can be seen in Fig. 5d and 5e, the damage of PC12 cells induced by DENP is alleviated with the aid of MOF-808-Al, which is in accordance with the descriptions in the manuscript.

Fig. 5. (a) Cell viability after treatment with different contents of MOF-808-Al for PC12 cell (Data are represented as mean \pm SD (n = 5), ns represented no statistical difference). (b) Relative activity of AChE in PC12 cells before and after treatment with DENP in the presence of various amounts of MOF-808-Al (Data are represented as mean \pm SD (n = 5), *p < 0.05, ***p < 0.001, ****p < 0.0001). (c) Fluorescence intensity of JC-1 (E_m: 595 nm) in different systems, including PC12, PC12 + DENP, PC12 + MOF-808-Al, and PC12 + MOF-808-Al + DENP, and (d, e) corresponding cell images showing fluorescence changing (Data are represented as mean \pm SD (n = 5), **p < 0.01, ****p < 0.0001).

Q2: Methods for evaluating the therapeutic effects *in vivo* should be described in paper body. In Fig 6c, the activity of AChE was detected in the blood from the mouse or other samples? More experimental evidence is required to validate the protective effect except HE staining analysis.

R2: Thanks for your valuable comment. The experiment procedure of therapeutic effects *in vivo*

was given in the revised manuscript.

The serum samples of mice were collected after different treatments for evaluating the AChE expression. Detailed information has been given in the experimental section.

“Protective effect of MOF-808-Al *in vivo*. The healthy mice were purchased from Hubei Provincial Center for Disease Control and Prevention. The *in vivo* animal experiments were performed according to the guidelines of the National Institutes of Health Guide for the Care and Use of Laboratory Animals and approved by the Ethics Committee of Medical College, Wuhan University of Science and Technology (2023082). The mice were raised with free access to standard feed and water in a 12 h dark-light cycle and the ambient conditions of room temperature (20 ~ 24 °C), 50 ± 5% relative humidity. A total of 16 mice were randomly assigned to 4 cages (n = 4 for every group). One cage of mice was used as a negative control and treated with only normal saline. Three cages of mice were separately treated with DENP (10 ug kg⁻¹), MOF-808-Al (2 mg kg⁻¹), and MOF-808-Al + DENP every day. After treatment for 8 days, all mice were anesthetized using diethyl ether. Blood was collected and centrifuged for 15 min at 3000 rpm to collect serum samples for further biochemical analysis. Tissues (brain, kidney, heart, and gut) from each mouse were dissected and immediately frozen in liquid nitrogen and stored at -80 °C.”

To further confirm the protective effect of the MOF-808-Al, the expression of alterations of biochemical markers was examined in mice and PC12 cells upon DENP exposure. Previous studies demonstrated that the organophosphorus compound can induce oxidative stress injury (*J. Hazard. Mater.* **2018**, 357, 348-354; *Food Chem. Toxicol.* **2022**, 169, 113432). Therefore, the content of reactive oxygen species (ROS) in DENP-treated PC12 cells was first determined by a common fluorescent probe (2',7'-dichlorodihydrofluorescein diacetate, DCFH-DA). As displayed in **Fig. R4**, the invasion of DENP significantly increases the level of ROS, which can further induce cell damage. Note that the content of ROS effectively decreases with the help of MOF-808-Al. Besides, the levels of three ROS-involved biomarkers, including superoxide dismutase (SOD), glutathione (GSH), and malonaldehyde (MDA), in the liver tissues of mice are monitored. The elevated MDA level and decreased expression of SOD and GSH induced by DENP are closely related to excessive ROS production (**Fig. R5**). Fortunately, with the aid of MOF-808-Al, the expressions of these biomarkers in DENP-treated tissues are the same as the normal tissues. Furthermore, the alkaline phosphatase (ALP) expression in serum, which is usually regarded to be associated with liver disorders, is tested. As shown in **Fig. R6**, the increased level of ALP conferred by DENP can be reduced as much as in healthy mice with the assistance of MOF-808-Al. It should be pointed out that the MOF-808-Al with good biosafety does not affect the expression of biochemical markers. Combing the HE staining analysis results, it is believed that the proposed MOF-808-Al exhibits a good protective effect to efficiently attenuate the damage of DENP.

Fig. R4. CLSM images of DCFH-DA-stained PC12 cells after treatment with DENP, MOF-808-Al, and MOF-808-Al + DENP.

Fig. R5. The level of (a) MDA, (b) GSH, and (c) SOD in liver tissues of mice after treatment with DENP, MOF-808-AI, and MOF-808-AI + DENP (Data are represented as mean \pm SD (n = 3), *p < 0.05, **p < 0.01, ***p < 0.001, ns represented no statistical difference).

Fig. R6. ALP activity in serum of mice after treatment with DENP, MOF-808-AI, and MOF-808-AI + DENP (Data are represented as mean \pm SD (n = 3), *p < 0.1, ns represented no statistical difference).

Q3: In addition to MTT assay, biocompatibility and safety evaluation of MOF-808-AI *in vivo* should be carried out.

R3: Thanks for your valuable comment. The cell live/dead double staining assay was conducted. The cell viability of PC12 cells after treatment with MOFs was unchanged in comparison with the control group, indicating good biocompatibility (**Fig. R7**). In addition, the favorable biosafety of MOF-808-AI *in vivo* has been confirmed by using the hematoxylin–eosin (HE) staining. As can be seen in **Fig. 6d**, Supplementary **Fig. 41**, the hippocampus, and main organs, including the heart, lung, liver, spleen, and kidney in the MOF-808-AI group show no observable abnormality. Furthermore, the MOF-808-AI treatment group exhibits no abnormal expression of biochemical markers in mice (**Fig. 6f-i**).

The relative expressions are updated in the revised manuscript for a comprehensive evaluation of the biosafety of the proposed MOF-808-AI.

Fig. R7. CLSM images of Calcein-AM/PI-stained cells treated with varying MOFs.

Fig. 6. (a) Schematic illustration of the OP-damaged model mice and the MOF-808-AI-mediated treatment. (b) Mice weight after treatment with saline, DENP, MOF-808-AI, and MOF-808-AI + DENP for 8 days (Data are represented as mean \pm SD (n = 4), **p < 0.01). (c) Relative activity of AChE in the serum of mice after various treatments. (Data are represented as mean \pm SD (n = 4), ****p < 0.0001, ns represented no statistical difference). H&E staining (d) hippocampus and (e) liver and kidney after different treatments. The level of (f) MDA, (g) GSH, (h) SOD in liver tissues of mice, and (i) the level of ALP in the serum of mice after different treatments (Data are represented as mean \pm SD (n = 3), *p < 0.05, **p < 0.01, ***p < 0.001, ns represented no statistical difference).

Supplementary Fig. 41. H&E staining of heart, lung, liver, spleen, and kidney of mice after treatment with DENP, MOF-808-AI, and MOF-808-AI + DENP.

Q4: There is no scale bar in Figure 1b, Fig 5d and 5e. Last but not least, a small spelling error in introduction (To alleviate the never damage) but there are several like this in the manuscript so the language needs carefully polishing.

R4: According to your suggestion, the scale bars were given. Also, grammar errors and typos were corrected carefully in the revised manuscript.

Reply to reviewer 3:

Reviewer 3: Xu *et al.* report a biomimetic heterogeneous MOF-808-Al catalyst with the ability to act like acetylcholinesterase, a critical enzyme required for the functioning of the nervous system. Importantly, this catalyst also decomposes neurotoxic organophosphate compounds. They show that the catalyst is therefore a good treatment for organophosphate (OP) poisoning in cell lines and in mice. This work is exciting and significant as while enzymes have been explored for OP poisoning, heterogeneous catalysts have not.

I feel that the work would be suitable for publication after major revisions addressing the below points:

Q1: The evidence for the catalytic mechanism of action and why the addition of Al³⁺ increases the activity of MOF-808 is not clearly presented. In Figure 4d, the authors present a reaction energy diagram to compare the energy profile of acetylcholine hydrolysis catalyzed by MOF-808 and MOF-808-Al. However, the authors only calculated the energies of the intermediate states, and not the transition states. The authors should clarify how they obtained energy barriers of 1.472 eV for the MOM-808 system and 1.068 eV for the MOM-808-Al system, as stated on page 8.

R1: Thanks for your valuable comment. Nature has produced a variety of enzymes that can rapidly hydrolyze substrates at active sites that contain Lewis acidic metal–oxy/hydroxy species, such as the Zn–OH and Zn–OH–Zn active centers in the enzyme of carbonic anhydrase and phosphotriesterase. In this context, researchers have leveraged the excellent chemical and thermal stability of Zr-based MOFs to develop robust heterogeneous catalysts to hydrolyze organophosphorus nerve agents (*Nat. Commun.* **2022**, *13*, 827; *J. Am. Chem. Soc.* **2021**, *143*, 18261-18271; *Nat. Mater.* **2015**, *14*, 512-516). The activities of these materials originate from the accessible Lewis acid Zr–O(H)–Zr groups and basic-nucleophilic O²⁻/OH⁻ sites, being capable of activating P–X (X = F, O, S) bonds. In details, the hard Lewis acid sites (Zr⁴⁺) can easily activate a phosphoryl group by accepting an electron lone pair from the oxygen and drawing electron density away from the double bond, leading to a greater positive charge on, and thus increasing the electrophilicity and reactivity of the central phosphorus. Subsequently, the cleavage of the P–X bond occurs under the nucleophilic attack of neighboring –OH. Then, the hydrolysis product is generated with the help of H₂O, accompanied by the regeneration of catalysts (**Fig. R8**).

Inspired by this, MOF-808, where the Zr₆ clusters have six coordinatively unsaturated Zr sites occupied by terminal –OH₂/OH groups, was selected as the ideal model in this work. The strong Lewis acidic Al³⁺ was decorated onto the Zr-oxo clusters to form the new and accessible active sites for promoting hydrolysis. First, more than 2 times higher Lewis acidity by Al³⁺ doping has been confirmed. Then, *operando* attenuated-total-reflection FTIR (ATR-FTIR) spectroscopy revealed that the Al³⁺ site provided a strong interaction force to induce the polarization of the carbonyl (C=O) or phosphoryl (P=O) bond. In addition, the consumption of –OH groups and the synergistic effect between the M–OH sites and Lewis acid sites has been verified. Based on these, we concluded that the exposed Al³⁺ or Zr⁴⁺ combines with oxygen of carbonyl groups to increase the electrophilicity and reactivity of carbon (phosphorous) atoms *via* the polarization effect. Subsequently, the nucleophilic attack of neighboring –OH induces the cleavage of the C–O (P–O) bond. For the third step, the intermediate is further hydrolyzed to the product in the presence of H₂O (Supplementary **Figs. 26** and **27**), being accompanied by the regeneration of –OH defects on the Al (Zr) sites. Furthermore, theoretical calculations reveal that the Lewis Al³⁺ sites have a stronger polarization effect than the Zr⁴⁺ sites, strengthening the electrophilicity and reactivity of C atoms. Assisted by the highly electronegative –OH groups, the proposed MOF-808-Al exhibits a decreased energy barrier for the dissociation of ester bonds and desorption of hydrolysates.

Fig. R8. The general mechanism for the catalytic hydrolysis of a phosphate bond (*Nat. Commun.* **2022**, *13*, 827).

According to your nice suggestion, the transition states (TS) search of nucleophilic attack of -OH* was carried out. As a result, the barrier for nucleophilic attack of Zr-OH* in the MOF-808 system is 0.609 eV (**Fig. R9**). As for the MOF-808-Al system, the highest energy change is the final state (0.912 eV), implying that the Al-OH* attack is controlled by thermodynamics. Although the Zr-OH* for nucleophilic attack is energetically favorable in comparison with the Al-OH*, the barrier is lower than that of the breaking of the C=O bond and the formation of choline, which is a rate-determining step (RDS). Compared with the MOF-808 system (1.472 eV), the lower energy barrier of RDS in the MOF-808-Al system (1.068 eV) reveals higher hydrolytic activity.

Fig. R9. The reaction energy diagram of MOFs for hydrolysis of ACh.

Additionally, the calculation of energy barriers is given in the experimental section.

“The energy change of the reaction is calculated as equations 1-3:

Q2: The authors postulate that the role of Al³⁺ is to act as a strong Lewis acid to activate acetylcholine, and to increase the nucleophilicity of OH*, which is required for attacking the conjugated acetylcholine to hydrolyze it. This is supported by Bader charge analyses showing a larger positive charge on the carbonyl C after conjugation to Al³⁺ than conjugation to Zr⁴⁺, and a larger negative charge on OH* groups attached to Al³⁺ than to Zr⁴⁺.

While these are reasonable arguments, they should be supported by transition state calculations to calculate the barriers of the attack of OH* on the carbonyl C, i.e. the ACh*→ACh-OH* step in Figure 4d. For MOF-808-Al, acetylcholine and OH* are both attached to the same Al, which means the transition state likely goes through a 4-membered ring state, which may be energetically

unfavorable. On the other hand, for MOF-808, acetylcholine and OH* are attached to different Zr centers, and so the TS may be energetically more favourable. For MOF-808-Al, the authors may also consider attack of a nearby Zr-OH* on the Al-ACh* complex, which may be more favorable.

R2: Thanks for your valuable comment. The barriers for the nucleophilic attack of OH* are performed by TS search. Although the Zr-OH* for nucleophilic attack exhibits energetically favorable in comparison with the Al-OH*, the barrier is lower than that of the RDS. Compared with that of the MOF-808, the lower energy of RDS in the MOF-808-Al system reveals the superior activity for AChE hydrolysis.

Besides, the reaction energies of the nearby Zr-OH* on the Al-ACh* complex are carried out. Different from the above systems, the RDS is the nucleophilic attack of Zr-OH*, which is calculated to be 1.230 eV (**Fig. R10**). The TS search indicates that the attack of Zr-OH* has thermodynamic rate control. This value is higher than that of the Al-OH* system (0.912 eV). This comparison makes it clear that the Zr-OH* attack pathway is energetically less favorable than the Al-OH* pathway. This result is agreed with the previously reported hydrolysis of the phosphoryl (P-O) bond (*J. Am. Chem. Soc.* **2023**, *145*, 7435–7445), where the defect monometallic M-OH serves as the catalytic site for substrates binding and nucleophilic attack, and showing in energetically favorable. Therefore, it is believed that the proposed mononuclear Al-OH sites show a better ability for hydrolysis in this work. The relative expression is updated in the revised manuscript.

“As can be seen in **Fig. 4d**, the barrier of nucleophilic attack of Zr-OH is calculated to be 0.455 eV, which is lower than that of the Al-OH (0.912 eV). Besides, the transition state search indicates that the energy of the Zr-OH attack is 0.605 eV. However, this process in the MOF-808-Al system is controlled by thermodynamics. Although the barrier of the Zr-OH* attack exhibits energetically favorable, the energy is lower than that of the breaking of the C=O bond and the formation of choline, which is the rate-determining step (RDS). Compared with the MOF-808 system (1.472 eV), the low energy barrier in the MOF-808-Al system (1.068 eV) reveals high hydrolytic activity. In addition, the possibility of adjacent Zr-OH* for the nucleophilic attack in the MOF-808-Al system was examined (**Supplementary Fig. 30**). The nucleophilic attack of Zr-OH* is RDS, which is calculated to be 1.23 eV. This process with higher energy is less favorable in comparison with Al-OH* attack. Therefore, it is believed that the carbon atom is most likely to be attacked by Al-OH* in the MOF-808-Al system.”

Fig. R10. The reaction energy diagram of MOF-808-Al, where the Al acts as a binding site, and Zr-OH acts as a nucleophilic attack group.

Q3: The authors point out that M-OH* groups are depleted during the reaction of MOF-808-Al with DENP (Figure 4b), which means that the reaction is not catalytic as the MOF catalyst will become deactivated after all M-OH* groups are consumed. Are M-OH* groups depleted during the reaction of the MOF catalysts with acetylcholine as well? If one OH* group can only react with one molecule of acetylcholine/DENP, would this mean that high doses of the catalyst would be needed to achieve protection/neutralization of OP poisoning, and would these doses be reasonable for typical poisoning cases in humans?

R3: Thanks for your valuable comment. To monitor the catalytic process, the *operando* attenuated-total-reflection FTIR (ATR-FTIR) spectroscopy was carried out. As shown in **Fig. R11**, the rapid decrease of peak M-OH ($\sim 3200\text{ cm}^{-1}$) was observed at the initial reaction. The consumption of the -OH groups is integral to the decomposition of OPs (*J. Phys. Chem. C* **2017**, *121*, 11261–11272). And then, the signal intensity of the -OH group is basically unchanged, implying that the consumption and regeneration of -OH groups reach equilibrium. The relative expression is updated in the revised manuscript.

The substrate hydrolysis cycle comprises four steps: (i) chemisorption of the substrate on the defect (active site) of the Al (Zr)-cluster node via C=O (P=O) moiety nucleophilic substitution; (ii) nucleophilic attack on the carbon (phosphorus) center of the chemisorbed substrates via an adjacent -OH, formation of a C-O-M (P-O-M) bridge between the substrate intermediate and Al (Zr) site; (iii) release of a product upon cleavage of the labile C-O (P-O) bond; (iv) the release of the hydrolysis product in the presence of H₂O, resulting in regeneration of -OH defects on the Al (Zr)-cluster node (Supplementary **Figs. 26** and **27**). In addition, this catalytic reaction is performed in the basic environment. The M-OH active sites can also be regenerated by proton exchanged with the solvent (*Chem* **2020**, *6*, 3118-3131). After cycle for several times, the obtained MOF-808-Al almost keeps the initial hydrolase-like activity (**Fig. 3e**), further confirming their good recyclability. Based on these, it is believed that the proposed mimics can well perform the protective effect *in vivo*.

Fig. R11. In-situ ATR-FTIR spectra of MOF-808-Al-induced DENP hydrolysis systems.

Minor points:

Q1: The Bader charge figure (Figure S29) is confusing as the numbers in the figures are the absolute number of valence electrons, instead of the charge, as is claimed in the caption.

R1: According to your nice suggestion, the related expression is updated in the revised manuscript.

Supplementary Fig. S31. The valence electrons of (a) MOF-808 and (b) MOF-808-Al after binding with ACh.

Q2: Could the authors explain why MOF-808 was chosen for functionalization, instead of simple ZrO₂ nanoparticles for example?

R2: Thanks for your valuable comment. Compared with the ZrO₂ nanoparticles, MOFs, comprising uniformly arrayed metal-containing nodes separated by organic linkers, feature higher functional tailorability, and larger surface areas porosity. It is beneficial for increasing the accessibility to the active catalytic sites and endowing MOFs with excellent catalytic ability (*J. Am. Chem. Soc.* **2021**, *143*, 18261-18271). In this regard, MOF-808 based on the tritopic benzene-1,3,5-tricarboxylate linker and 6 connectivity, exhibiting a hierarchical porous structure, is the ideal model. Importantly, the Zr₆O clusters have six coordinatively unsaturated Zr sites occupied by terminal -OH₂/OH groups, which can serve as active sites for hydrolysis.

Q3: The authors should explain how the “Free Energies” in Figure 4d were obtained—were frequency calculations also performed with VASP?

R3: Thanks for your valuable comment. The energy change of the reaction is calculated as equations 1-3:

And the corrected expression (**Fig. 4b**) is updated in the revised manuscript.

Reviewers' Comments:

Reviewer #1:

Remarks to the Author:

The authors have addressed the reviewer comments which included carrying out additional experiments. I feel that the manuscript is suitable for being published.

Reviewer #2:

Remarks to the Author:

The authors have addressed my previous concerns, performed additional work and revised the manuscript accordingly. Thus, I recommend its publication in Nature Communications.

Reviewer #3:

Remarks to the Author:

The authors have addressed most of my concerns adequately. However, there is one point that remains of concern:

1) The authors seem to use the term "barriers" to describe the energy of reaction, for example, in R1 and in the main text, the authors state that:

"Compared with the MOF-808 system (1.472 eV), the lower energy barrier of RDS in the MOF-808-Al system (1.068 eV) reveals higher hydrolytic activity. Fig. R9"

These values are the energies of reaction according to Figure 4d. However, barriers typically refer to activation energies that can only be obtained by transition state calculations.

In accordance to my previous comment, the authors have performed one transition state calculation. However, for the rest of the reactions, the energies of reactions are still being referred as "barriers". The authors are urged to perform transition state calculations for all of the reactions, or reformulate their argument based on energies of reaction instead of barriers.

Reply to reviewer 1:

The authors have addressed the reviewer comments which included carrying out additional experiments. I feel that the manuscript is suitable for being published.

R1: Thanks for your valuable comment.

Reply to reviewer 2:

The authors have addressed my previous concerns, performed additional work and revised the manuscript accordingly. Thus, I recommend its publication in Nature Communications.

R1: Thanks for your valuable comment.

Reply to reviewer 3:

The authors have addressed most of my concerns adequately. However, there is one point that remains of concern:

Q1: The authors seem to use the term “barriers” to describe the energy of reaction, for example, in R1 and in the main text, the authors state that:

“Compared with the MOF-808 system (1.472 eV), the lower energy barrier of RDS in the MOF-808-Al system (1.068 eV) reveals higher hydrolytic activity. Fig. R9”

These values are the energies of reaction according to Figure 4d. However, barriers typically refer to activation energies that can only be obtained by transition state calculations.

In accordance to my previous comment, the authors have performed one transition state calculation. However, for the rest of the reactions, the energies of reactions are still being referred as “barriers”. The authors are urged to perform transition state calculations for all of the reactions, or reformulate their argument based on energies of reaction instead of barriers.

R1: Thanks for your valuable comment. According to your nice suggestion, we have reformulated the argument based on energies of reaction instead of barriers. The relevant expressions have been updated in the revised manuscript. Please see the yellow marked parts in the revised manuscript.

Reviewers' Comments:

Reviewer #3:

Remarks to the Author:

The authors have addressed my concerns adequately and I believe the manuscript is now suitable for publication.

Reply to reviewer 3:

The authors have addressed my concerns adequately and I believe the manuscript is now suitable for publication.

R1: Thanks for your valuable comment.